# Cellular Transcriptomics of Carboplatin Resistance in a Metastatic Canine Osteosarcoma Cell Line

**DOI:** 10.3390/genes14030558

**Published:** 2023-02-23

**Authors:** McKaela A. Hodge, Tasha Miller, Marcus A. Weinman, Brandan Wustefeld-Janssens, Shay Bracha, Brian W. Davis

**Affiliations:** 1Department of Veterinary Integrative Biosciences, College of Veterinary Medicine and Biomedical Science, Texas A&M University, College Station, TX 77840, USA; 2Department of Small Animal Clinical Sciences, College of Veterinary Medicine and Biomedical Science, Texas A&M University, College Station, TX 77840, USA; 3Department of Cellular, Molecular and Biomedical Sciences, The University of Vermont, Burlington, VT 05405, USA; 4Department of Clinical Sciences, College of Veterinary Medicine Flint Animal Cancer Center, Colorado State University, Fort Collins, CO 80523, USA; 5Department of Clinical Sciences, College of Veterinary Medicine, The Ohio State University, Columbus, OH 43210, USA

**Keywords:** osteosarcoma, chemoresistance, tumor trajectory, EMT, tumor microenvironment

## Abstract

Osteosarcoma prognosis has remained unchanged for the past three decades. In both humans and canines, treatment is limited to excision, radiation, and chemotherapy. Chemoresistance is the primary cause of treatment failure, and the trajectory of tumor evolution while under selective pressure from treatment is thought to be the major contributing factor in both species. We sought to understand the nature of platinum-based chemotherapy resistance by investigating cells that were subjected to repeated treatment and recovery cycles with increased carboplatin concentrations. Three HMPOS-derived cell lines, two resistant and one naïve, underwent single-cell RNA sequencing to examine transcriptomic perturbation and identify pathways leading to resistance and phenotypic changes. We identified the mechanisms of acquired chemoresistance and inferred the induced cellular trajectory that evolved with repeated exposure. The gene expression patterns indicated that acquired chemoresistance was strongly associated with a process similar to epithelial–mesenchymal transition (EMT), a phenomenon associated with the acquisition of migratory and invasive properties associated with metastatic disease. We conclude that the observed trajectory of tumor adaptability is directly correlated with chemoresistance and the phase of the EMT-like phenotype is directly affected by the level of chemoresistance. We infer that the EMT-like phenotype is a critical component of tumor evolution under treatment pressure and is vital to understanding the mechanisms of chemoresistance and to improving osteosarcoma prognosis.

## 1. Introduction

Osteosarcoma (OSA) treatment and prognosis have remained unchanged over the past thirty years [1]. It continues to be the most common human primary bone malignancy and comprises under 1% of adult cancers and 3–5% of pediatric cancers [2]. The current standard of care, first implemented in the 1980s, entails surgical excision, radiation, and an aggressive chemotherapy regimen [1,3]. This offers a ten-year survival rate of 60% for patients with nonmetastatic disease and 30% for patients with metastatic disease [1,3].

Canine OSA is a naturally occurring cancer and the predominant bone malignancy that affects dogs. Most dogs impacted by OSA are large or/giant breeds that experience increased cell proliferation and rapid growth in the long bones of the appendicular skeleton [4]. In addition, the rate of incidence is 10-fold greater in dogs than that in humans, and a canine’s shorter lifespan allows for more rapid data collection [4]. The canine model is particularly relevant due to mutations in the *TP53*, *RB1*, *MTAP/CDKN2A*, and *MDM2* genes that are commonly associated with human and canine OSA [5,6,7]. However, with the exception of *TP53*, the specific mutations within these shared genes that infrequently occur in both species and mutations in *SETD2* and *DMD* are unique to canine OSA [6,8,9,10].

The primary cause of OSA treatment failure is chemoresistance, which has been attributed to tumor adaptability and the trajectory of tumor cell evolution [11,12]. Tumor adaptability is driven by the key hallmarks of tumorigenesis, genomic instability, and compromised regulation of DNA replication and the cell cycle [11]. Genomic instability results in high levels of copy number variation and genotypic, transcriptomic, and phenotypic heterogeneity within cancerous tissues that can promote adaptive mechanisms for resistance to chemotherapy drugs and the production of highly resistant cancer cells [11,12]. In addition, chemotherapy-induced mutations may drive the tumor’s trajectory by selecting for drug-resistant cell populations [11,12,13].

The gold-standard chemotherapies for OSA in both humans and canines are platinum-based drugs [14]. The mechanisms of resistance are similar among platinum chemotherapeutic agents and include reduced drug accumulation, inactivation by glutathione (GSH) and metallothionein (MT), increased DNA repair, and suppression of apoptosis [13,14,15,16].

A preliminary study characterized the proteomes of canine osteosarcoma HMPOS-derived carboplatin-resistant cell lines and their exosomes, and evaluated exosome-mediated chemotherapy resistance [17]. There were differences between each cell line in the response to chemotherapeutics and in the proteins expressed [17]. Proteomic analysis indicated an association of chemotherapy resistance with the glutathione biosynthesis, conjugation, and recycling pathways, and the γ-glutamyl biosynthesis pathway [17]. These pathways minimize the effectiveness of chemotherapeutic drugs by promoting glutathione S transferase (GST) enzyme activation, which hydrolyzes the platinum agents’ active group [17,18]. Carboplatin-resistant cell lines, particularly HMPOS-10R, have high expression of β-catenin, an oncogene [17]. β-catenin plays a key role in the Wnt/β-catenin signaling pathway and promotes chemoresistance through upregulating *MDR1* and inducing epithelial–mesenchymal transition (EMT) [19]. Exposure to HMPOS-2.5R exosomes was found to induce chemotherapy resistance in naïve HMPOS-S cells, while exposure to HMPOS-10R exosomes only induced β-catenin expression in naïve HMPOS-S cells [17]. The results of this study indicated the complexity of chemotherapy resistance induction and conveyance, which we will continue to investigate by evaluating the transcriptomes of these cell lines [17].

We hypothesized that OSA cancer cells develop their drug resistance by adapting to the evolutionary selective pressure resulting from increased chemotherapy concentration, either along a singular or a diversified path. The goal of this study was to examine the evolutionary trajectory, assess the results of adaptation due to selective pressure, and determine the nature of resistance mechanisms in previously induced OSA carboplatin-resistant cell lines. This in vitro approach mitigates the cellular complexity of the bone marrow niche to address the diverse tumor microenvironment, which aids in tracking the evolutionary progress of each cell population as they acquire carboplatin resistance [20].

## 2. Materials and Methods

### 2.1. Carboplatin Chemoresistant Cell Lines

For our experiment, we used a highly aggressive and malignant cell line, Highly Metastasizing POS (HMPOS). The HMPOS cells were a kind donation from the Barroga and Fujina Lab [21]. This cell line was previously derived from the canine OSA cell line POS, which was generated by harvesting cells from canine metastatic lesions passaged in mice. Cells were incubated at 37 °C with 5% CO_2_ in growth media (RPMI 1640 and 10% FBS supplemented with 100 units/mL penicillin and 100 μg/mL streptomycin) [21]. All cells were tested for mycoplasma prior to incubation. The cells used for this study were passaged fewer than five times before arriving in our lab. Expansion in-house entailed fewer than 3 passages prior to introducing carboplatin. Carboplatin resistance was induced at 0, 2.5 μM, and 10 μM dosages in a previous study, and clones were validated to be resistant and were used in this study (Figure 1) [21]. The HMPOS 0, 2.5 μM, and 10 μM-carboplatin-resistant cells will be referred to as HMPOS, HMPOS-2.5, and HMPOS-10, respectively.

### 2.2. Carboplatin Sensitivity Assay

HMPOS, HMPOS-2.5, and HMPOS-10 cell lines were grown in T-25 tissue culture flasks containing RPMI 1640 cell culture media with 10% fetal bovine serum. The flasks were incubated at 37 °C with 5% CO_2_ until 50% confluent. The tissue culture flasks were washed with 1× PBS and replenished with serum-free RPMI 1640 cell culture media. The cells were incubated at 37 °C and serum-starved for 24 h. After incubation, tissue culture flasks were washed with 1× PBS, and varying concentrations of carboplatin diluted in cell culture media were added to the flasks. All three cell lines were treated with carboplatin doses ranging from 0–480 µM. Conditions were set up in triplicate and cells containing drug-free RMPI 1640 with 10% fetal bovine serum were used as a negative control. All flasks were incubated for 72 h at 37 °C. After incubation, the cells were trypsinized, harvested, and centrifuged at 500× *g* for 5 min. The supernatant was decanted and the remaining cell pellet was resuspended in 100 µL of 1× PBS. The cell suspension was mixed 1:1 with ViaStain AOPI Staining Solution (Nexcelom Biosciences, Lawrence, MA, USA) and cell viability was determined using a Cellometer Auto 2000 (Nexcelom Biosciences, Appendix A). Three technical replicates of three biological replicates were performed. IC50 values were determined using non-linear regression and curve fit analysis. To determine the difference between curves, pairwise extra sum-of-squares F tests were performed, with the null hypothesis as IC50 being the same and the alternative hypothesis as IC50 being different. We also tested this as IC50 being different in at least one of the three replicates. For all tests, *p* < 0.001.

### 2.3. Cell Invasion and Migration Assay

The coating buffer was produced from 0.7% NaCl and 0.1 M tris in distilled water and filtered using a 0.2 µm sterile syringe filter. The Matrigel Matrix (Corning, Tewksbury, MA, USA) was diluted using chilled coating buffer to a final concentration of 250 µg/mL. Thincerts were placed in a 24-well plate (Greiner Bio-one, Kremsmünster, Austria) and coated with 100 µL of Matrigel Matrix. The plates were placed in an incubator at 37 °C for 2 h. A cell suspension was prepared using serum-free cell culture media at a concentration of 1 × 10^6^ cells/mL. An amount of 600 uL of cell culture media containing serum was pipetted into the bottom of each test well and a coated Thincert was placed on top of the media; 200 µL of the cell suspension was pipetted into each Thincert and the plates were returned to the incubator. Positive control wells to measure migration were set up in the same manner as the test wells, but included uncoated Thincerts. Negative control wells contained uncoated Thincerts to measure invasion, but only contained serum-free media. All conditions were run in triplicate. Plates were removed after 20 h and the Thincerts were gently removed and placed in a new 24-well plate containing serum-free media and 8 µM Calcein-AM. The plates were incubated for 45 min at 37 °C. After incubation, the cell suspension was removed from the inside of the Thincert, and the Thincerts were placed in a new 24-well plate containing 500 µL of pre-warmed trypsin–EDTA. The plates were incubated for 15 min at 37 °C with occasional tapping to encourage detachment. After 15 min, the thincerts were discarded and the trypsinized cells were mixed via a pipette. A total of 200 µL of the trypsinized cells was pipetted into a 96-well plate in triplicate and the plates were read by a fluorescent plate reader (excitation 485 nm/emission 520 nm) (BioTek Synergy 2, Winooski, VT, USA). The invasion index percentage was calculated using the following formula: Invasion index % = (experimental average–negative control average)/(positive control average–negative control average) × 100%. The migration index was calculated using the following formula: Migration index% = (positive control average − negative control average)/200,000 (total number of cells that were originally plated). Analysis was performed using ANOVA single-factor analysis to evaluate the *p* values for the invasion and migration assays, and Tukey’s HSD test was performed for the invasion assay data (Appendix A).

### 2.4. Single Cell RNA Sequencing

Cell suspensions were prepared using Next Gel Bead-in-Emulsion (GEM) technology using the Chromium Controller (10× Genomics). The Chromium Single Cell 3′ Library, Gel Bead & Multiplex Kit v3 (10× Genomics) was used to construct the scRNA-seq libraries following the manufacturer’s protocols. Sequencing libraries were constructed using the Nextera XT DNA sample Pre-Kit (Illumina, San Diego, CA, USA). The final libraries were analyzed using the Agilent Bioanalyzer by running a High Sensitivity DNA Kit (Agilent Technologies, Santa Clara, CA, USA). The pooling of individual libraries was performed using 75-cycle run kits on an Illumina HiSeq X platform with 150 bp paired-end reads. The Texas A&M Institute for Genome Sciences and Society performed the scRNA sequencing (Figure 1).

### 2.5. Analysis and Visualization

All following computational methods were performed on a private 96-core server running Scientific Linux v7. Raw base call files (BCL) were demultiplexed using the cellranger mkfastq command to generate FASTQ files [22]. These FASTQ files were filtered and the cell barcodes and unique molecular identifiers (UMIs) were extracted using Cell Ranger v6.0 and then aligned to the canFam4 reference [23]. The “count” command then was used to group reads with the same cell barcodes, UMIs, and genes to calculate the number of UMIs per gene per cell. The Seurat package (version 4.0.0) was used to process the raw output data in R Studio software (version 3.3) for each individual tissue sampled [24,25]. Cells were filtered for bioinformatic analysis by parameters to exclude cells with the percentage of mitochondrial genes expressed over 5% and those that fell outside of the 250–2500 number of genes. The filtered cells underwent cell cycle regression to minimize cell cycle heterogeneity effects. The filtered cells were then scaled and underwent principal component analysis (PCA), nearest neighbors were computed, and clusters were found using a resolution of 0.5. The data were dimensionally reduced using both t-distributed Stochastic Neighbor Embedding (tSNE) and Uniform Manifold Approximation and Projection (UMAP). Differential expression was determined using the R package Seurat between each cell line, and genes with a *p*-value less than 2 × 10^5^ were further evaluated in gene enrichment and network analyses. Pseudotime analysis was conducted using Monocle and Slingshot [24,26,27].

Each cell line was mostly homogenous, but scRNAseq was performed in order to capture small heterogeneous populations that illustrated the evolutionary trajectory that bulk RNA sequencing would not be able to detect. Differential expression testing was performed by pairwise comparison of each cell line (Appendix A), between each line and all other cells (Appendix A), between the three Seurat clusters within HMPOS-10 (Appendix A), and between clusters 8, 10, and 11 with their respective primary cell line clusters HMPOS, HMPOS-10, and HMPOS-10, respectively (Appendix A). Gene set enrichment analysis was performed on the upregulated and downregulated gene lists, comparing HMPOS with HMPOS-2.5, HMPOS with HMPOS-10, and HMPOS-2.5 with HMPOS-10 in gProfiler [28]. Network and pathway analyses were performed on all differential expression comparisons with the core analysis algorithm in Ingenuity Pathway Analysis (IPA) to compare the log fold-change (LogFC) variation for significant genes and the Bonferroni-corrected *p*-values within each gene list [29]. A visual overview of this experiment illustrates the pipeline from cell culture to sequence data to cell analysis (Figure 1).

**Figure 1 genes-14-00558-f001:**
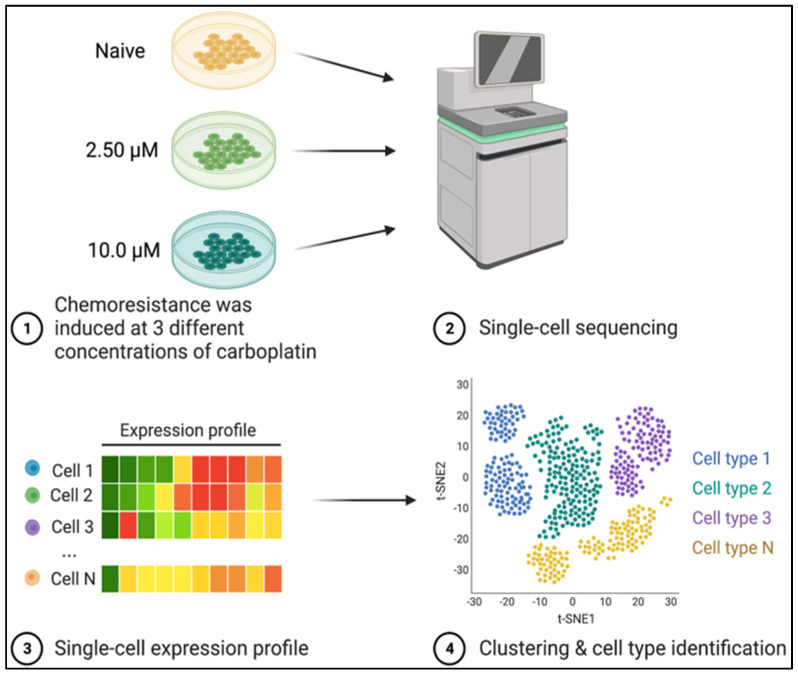
Visual overview of experimental design. HMPOS cell lines were previously challenged with increased concentrations of carboplatin to generate the HMPOS-2.5 μM and HMPOS-10 μM drug-resistant cell lines [23]. Single-cell RNA sequencing was performed on these two chemoresistant cell lines and the HMPOS cells. Created with BioRender.com, (accessed on 22 October, 2022) [30].

## 3. Results

### 3.1. Characterization Confirmation of the HMPOS Cell Lines

The resistance of HMPOS, HMPOS-2.5, and HMPOS-10 confirmed the results of previous carboplatin sensitivity assay characterization (Figure 2A) [17]. The HMPOS-10 cell line exhibited the highest resistance to carboplatin, followed by the HMPOS-2.5 cell line, and the HMPOS cell line was found to be the least resistant (Figure 2A). The IC50-values calculated for HMPOS, HMPOS-2.5, and HMPOS-10 were 88.10 μM, 151.99 μM, and 248.85 μM, respectively. There was no statistical difference between each group using the extra sum-of-squares F test with *p*-values of 0.653, 0.368, and 0.377 for HMPOS and HMPOS 2.5, HMPOS and HMPOS-10, and HMPOS-2.5 and HMPOS-10, respectively.

Cell morphology was visually different between the HMPOS cells, which exhibited a more cuboidal shape than the HMPOS-2.5 and 10 cells, which were comprised of a spindled morphology with a more severe shape in HMPOS-10 (Figure 2D–F). The Invasion Index Assay was statistically significant according to ANOVA analysis, but Tukey’s range test showed no statistical significance between samples, except for HMPOS-2.5 (Figure 2B). The Invasion Index assay indicated that HMPOS-10 was more invasive than HMPOS, but HMPOS-2.5 had the lowest invasion index scores of the three cell lines (Figure 2B). There was no statistically significant difference in the rate of migration between all three cell lines (Figure 2C).

### 3.2. Distinction in Transcriptomes between Chemoresistance Levels

Post-filtration, there were 12,644, 10,883, and 8668 cells for HMPOS, HMPOS-2.5, and HMPOS-10 respectively. The distribution across cell phases for each cell type was similar, with the exception of a lower proportion of cells in G1 in HMPOS-10 (Appendix A). Each cell line clustered distinctly using Principal Component Analysis (PCA) and Uniform Manifold Approximation and Projection (UMAP), resulting in fourteen subclone groups (Figure 3A). The inferred pseudotime values showed a trajectory in evolution from HMPOS to HMPOS-2.5, then HMPOS-10 (Figure 3C and Appendix A). A clear distinction between these three cell lines due to the differential expression was evident. In total, 688 genes were upregulated and 400 genes were downregulated in the HMPOS line in comparison with the HMPOS-2.5 cells; 1147 genes were upregulated and 1164 genes were downregulated in the HMPOS cells compared with the HMPOS-10 cells; and 976 genes were upregulated and 1465 genes were downregulated in the HMPOS-2.5 cells compared with the HMPOS-10 cells.

### 3.3. Differential Gene Expression in Chemoresistant Cell Lines

Genes that were differentially upregulated in HMPOS compared with the induced carboplatin-resistant cell lines were found to be related to biological pathways that involved biosynthesis, metabolism, and proliferation (Figure 4A). The biological pathways upregulated in HMPOS-2.5 cells from HMPOS were in response to chemicals and morphogenesis. Catalytic activity and protein binding were the upregulated biological pathways in HMPOS-2.5 compared with HMPOS-10 cells. Protein, enzyme, and transcription factor binding were upregulated in HMPOS-10 cells compared with the naïve and HMPOS-2.5 cell lines. Further analysis of pathway differentiation using IPA implicated the EIF2 signaling pathway, the sirtuin signaling pathway, and the mTOR signaling pathway. The EIF2 signaling pathway was activated in HMPOS-2.5 compared with HMPOS, while the sirtuin and mTOR pathways were downregulated in HMPOS-2.5 compared with HMPOS (Figure 5B). In HMPOS-10, the EIF2 signaling pathway was also activated, and the mTOR pathway was downregulated when compared with HMPOS and HMPOS-2.5 (Figure 5C). In contrast, the sirtuin pathway was activated in HMPOS-10 (Figure 5C). *TMSB4X*, *LGALS3, COL3A1*, and *HMGA1* increased and *THBS2, BAG2*, *BASP1*, *PEG10*, and *NPR3* decreased in correlation with the increase in the carboplatin dosage (Appendix A). The HMPOS-10 cell line was particularly distinct and exhibited high expression of *S100A6*, *TPM2*, *CCND2*, *TFPI2*, *HMGA2,* and *GSTT2B* (Appendix A). *CDKN2A*, *TMEM126A*, *TMEM126B, MTAP*, *COX7A1*, and *LTBR* were downregulated in the HMPOS-10 cell line (Appendix A). For the HMPOS-2.5 cell line, *PIEZO2*, *IGFBP2*, and *ACVR2A* were distinctly downregulated and *THBS1*, *SFRP2*, *PTN*, *FSIP1*, *DPT*, *OMD*, *OGN*, *COL5A2*, and *ENO1* were distinctly upregulated (Appendix A). Network and pathway analyses comparing HMPOS with HMPOS-2.5, HMPOS with HMPOS-10, and HMPOS-2.5 with HMPOS-10 are presented in Appendix A respectively.

### 3.4. Subclonal Cell Populations

For each cell line, there was a cluster distinct from the main cell population, only detectable using the single cell approach. Cluster 8 and cluster 13 were secondary clusters for HMPOS and were labeled as HMPOS-Var-1 and HMPOS-Var-2, respectively. HMPOS-Var-2 was not investigated further as it consisted of only 249 cells (Appendix A). Cluster 10 was a secondary cluster for HMPOS-2.5 and was labeled HMPOS-2.5-Var (Appendix A). Cluster 11 was a secondary cluster for HMPOS-10 and was labeled HMPOS-10-Var (Appendix A). Additionally, clusters 1 and 5 within the HMPOS-10 group were compared due to their divergent expression patterns. A comparison of HMPOS with HMPOS-Var-1 revealed the downregulation of *THBS1* and *LGALS3*. In HMPOS-2.5-Var, *FSIP1* and OMD were upregulated in comparison with HMPOS-2.5. SFRP2 and *COL12A1* were downregulated and *FSIP1* was upregulated in HMPOS-10-Var in comparison with HMPOS-10. Clusters 1, 5, and 9 within the main HMPOS-10 cell population were also compared using IPA and revealed the inhibition of the EIF2 signaling pathway and promotion of the sirtuin signaling pathway and the mTOR signaling pathway (Figure 5D and Appendix A).

## 4. Discussion

One of the main causes of treatment failure in OSA and many other cancers is chemotherapy drug resistance [11,12]. Tumor adaptability and the trajectory of cancer cell evolution play a major role in treatment evasion and the development of evasion mechanisms. It is essential to understand the trajectory of tumor adaptation in order to tailor treatment strategies to fit the genetic profile of patients in order to improve prognosis.

The cell viability assay confirmed the induction of carboplatin drug resistance, with HMPOS being the most sensitive, HMPOS-2.5 having improved survival, and HMPOS-10 exhibiting the highest resistance to carboplatin exposure. The observed morphology changed from cuboidal to spindled, correlating with the morphological changes seen in the epithelial-to-mesenchymal transition (EMT), where spindled tumor cells are more aggressive and chemoresistant [30]. The invasion and migration assays, which assessed the metastatic potential of cells via movement, were not significant between the three conditions. This was expected, as the metastatic nature of HMPOS was well established, and was not likely to change given exposure to carboplatin. However, the gene expression patterns between HMPOS, HMPOS-2.5, and HMPOS-10 were distinct from one another, with HMPOS-10 being the most phenotypically and transcriptionally divergent.

### 4.1. Epithelial-to-Mesenchymal Transition in Correlation with Chemotherapy Resistance

The resistance to carboplatin and change to a spindled morphology from HMPOS to HMPOS-2.5 to HMPOS-10 may indicate the induction of EMT or an EMT-like process as a result of adaptive resistance. Several lines of evidence for the resistant cell lines in this work demonstrate an evolutionary progression toward this transition. As chemoresistance increased, *TMSB4X*, *LGALS3, COL3A1*, and *HMGA1* were consistently upregulated, while *THBS2*, *BASP1*, *NPR3, BAG2*, and *PEG10* were downregulated.

The upregulation of *TMSB4X* (thymosin-β_4_ X-linked), *LGALS3* (galectin 3), *COL3A1* (collagen type III α 1), and *HMGA1* (high-mobility group AT-hook 1) is associated with chemoresistance and poor prognosis [31,32,33,34,35,36,37,38,39,40]. thymosin-β_4_ is an actin-binding protein and plays a major role in the development of tissue and wound repair [32,33]. The upregulation of *TMSB4X* is correlated with tumor progression and induces the activation of myocardin-related transcription factors (MRTF) that regulate EMT transition and downregulate E-cadherin [31,32,33]. *LGALS3* plays a role in apoptosis and cell adhesion, and stimulates bone marrow mesenchymal stem cells to express interleukin-6 (IL-6) to promote tumorigenesis, inflammation of the tumor microenvironment, and metastasis [34,35,36,37,38]. *TMSB4X* suppresses and *LGALS3* interacts with E-cadherin, a tumor-suppressor protein that maintains cell adhesion and epithelial structural integrity, and is a key gene that is downregulated to allow EMT transition [31,35]. Collagen type III α 1 is a component of the extracellular matrix [39,40]. *COL3A1* is a marker for EMT and is shown to be associated with *POSTN* (periostin), which activates the ERK and p38 pathways and downregulates miR-381 expression to regulate EMT [39,40]. *HMGA1* is an architectural transcription factor, and its overexpression activates Akt signaling to promote survival and proliferation [41,42,43]. The HMGA1–TRIP13 axis has been shown to induce EMT when HMGA1 is overexpressed [44].

*THBS2* (thrombospondin 2), *BASP1* (brain acid soluble protein 1), and *NPR3* (natriuretic peptide receptor 3) are tumor-suppressor genes, for which downregulation is associated with poor prognosis [45,46,47,48,49]. Deficiency of *THBS2* is associated with the degradation of collagen and the extracellular matrix to allow metastasis [45]. Upregulated miR-191 expression promotes EMT and activates the Wnt pathway for tumor promotion through the inhibition of *BASP1* [47,48]. *POU2F1* regulates *NPR3* expression to block the PI3K/Akt pathway to inhibit OSA cell proliferation and EMT [49].

*BAG2* (BAG cochaperone 2) plays a role in the Akt/mTOR and ERK pathways to promote tumorigenesis [50,51,52]. *PEG10* (paternally expressed gene 10) promotes tumor invasion and metastasis, and is a major regulator in TGFB1-induced EMT [53,54,55]. The increased chemoresistance, the cell morphology change, and the gene expression suggest that the gain of chemoresistance is associated with the transition from epithelial to mesenchymal. The downregulation of *BAG2* and *PEG10* with the increase in chemoresistance may indicate the effectiveness of other mechanisms in promoting tumorigenesis and resistance.

### 4.2. Epithelial-to-Mesenchymal Transition in HMPOS-10

HMPOS-10 exhibited high expression of genes associated with tumorigenesis and directly correlated with chemotherapy resistance, including *S100A6*, *TPM2*, *CCND2*, *TFPI2*, *HMGA2,* and *GSTT2B*. *S100A6* (calcium-binding protein A6) is upregulated in breast cancer through mesenchymal stem cell-secreted exosomes to promote chemotherapy resistance [56,57]. *S100A6* is involved in the Wnt/β-catenin signaling pathway and induces EMT by downregulating E-cadherin [56,57]. *TPM2* (tropomyosin 2) is mainly expressed in muscle fibers and, when upregulated, decreases E-cadherin and β-catenin expression [58,59]. *CCND2* (cyclin D2) is a driver of cell cycle progression and can be suppressed by miR-646 to prevent tumorigenesis and EMT [60,61,62]. *TFPI2* (tissue factor pathway inhibitor-2) is a tumor-suppressor gene that induces apoptosis, but hypermethylated *TFPI2* is associated with several human cancers and dysregulated *TFPI2* overexpression promotes EMT through the TGF-β pathway [63,64]. *HMGA2* (high-motility group AT-hook 2) overexpression activates the Dvl2/Wnt pathway to increase chemoresistance and promote EMT through the MAPK pathway [65,66]. *GSTT2B* (glutathione S-transferase theta 2) is a pseudogene of *GSTT2*, and glutathione S-transferases are associated with chemotherapy, such as platinum agents and detoxification [67].

The genes downregulated in HMPOS-10 were *CDKN2A*, *TMEM126A*, *TMEM126B, MTAP*, *COX7A1*, and *LTBR. CDKN2A* (cyclin-dependent kinase inhibitor 2A) is an inhibitor of cellular proliferation through the Akt/mTOR pathway, and loss-of-function correlates with chemotherapy resistance [68,69]. *TMEM126A* and *TMEM126B* (transmembrane protein 126A and transmembrane protein 126B, respectively) downregulation promotes mitochondrial and extracellular matrix dysregulation, attributed to poor prognosis, EMT, and chemoresistance [70]. EMT is also promoted by the purine metabolic enzyme *MTAP* (methylthioadenosine phosphorylase), which is downregulated in lung adenocarcinoma and predicts prognosis [71,72]. Knockout of *MTAP* was found to downregulate E-cadherin and *p*-GSK3β and lead to EMT progression [73]. *COX7A1* (cytochrome c oxidase subunit 7A1) is involved in the mitochondrial respiratory chain and its overexpression inhibits cell proliferation and promotes apoptosis [74]. *LTBR* (lymphotoxin β receptor) mediates apoptosis in tumor cells and activates tumorigenesis by promoting the NF-Kβ pathway [75,76].

Analysis of HMPOS-10 showed that there were many different genes at play targeting various pathways, including the Wnt/β-catenin, TGF-β, Dvl2/Wnt, MAPK, Akt/mTOR, and NF-Kβ pathways, which are related to tumorigenesis and the induction of an EMT-like phenotype. The mechanisms of chemotherapy resistance linked to EMT include improved proliferation and maintenance, resistance to apoptosis, the overexpression of ABC transporters that remove chemotherapeutics, and the induction of a hypoxic tumor microenvironment.

### 4.3. Epithelial-to-Mesenchymal Transition in HMPOS-2.5

HMPOS-2.5 exhibited upregulation of *THBS1*, *SFRP2*, *PTN*, *FSIP1*, *DPT*, *OMD*, *OGN*, *COL5A2*, and *ENO1* and downregulation of *PIEZO2*, *IGFBP2*, and *ACVR2A*. *THBS1* (thrombospondin 1) upregulation activates the TGF-β pathway to promote tumorigenesis and EMT [77,78]. *SFRP2* (secreted frizzled-related protein 2) modulates the Wnt/β-catenin pathway and controls *WNT16B* to promote acquired resistance [79,80]. In vitro and in vivo studies show that the downregulation of *SFRP2* expression can reverse the EMT process [81]. *PTN* (pleiotrophin) is a growth factor involved in proliferation and in osteosarcoma; its overexpression promotes EMT and doxorubicin resistance [82,83,84]. The upregulation of *FSIP1* (fibrous sheath interacting protein 1) correlates with poor prognosis in breast cancer and *FSIP1* knockout in a mouse model has been found to improve docetaxel sensitivity [85]. *DPT* (dermatopontin) promotes cellular adhesion and enhances *TGFB1* during the process of wound repair [86,87]. *OMD* (osteomodulin) is involved in osteoblast differentiation and *OGN* (osteoglycin) regulates bone and glucose homeostasis and has been indicated as a tumor suppressor [88,89]. In colorectal cancer, OGN upregulation has been found to induce EGFR endocytosis and inhibit EMT through the EGFR/Akt pathway [90]. EMT is accelerated and metastasis is promoted by the upregulation of *COL5A2* (collagen type V α 2 chain), while its downregulation inhibits the *TGF-*β and Wnt/β-catenin signaling pathways in OSA [91,92]. *ENO1* (enolase 1) is a glycolytic enzyme that suppresses ERK1/2 phosphorylation to inhibit EMT in vivo [93,94,95].

*PIEZO2*, *IGFBP2*, and *ACVR2A* upregulation is associated with the promotion of EMT [96,97,98,99,100,101]. *PIEZO2* (piezo-type mechanosensitive ion channel component 2) regulates the actin cytoskeleton, and actin remodeling can alter drug response [96,97]. *IGFBP2* (insulin-like growth factor-binding protein 2) promotes cellular proliferation, and its upregulation correlates with chemotherapy resistance [98,99,100]. *ACVR2A* (activin A receptor type 2A) mediates members of the TGF-β family and its loss-of-function results in an increase in tumorigenesis and metastasis [101].

The difference in gene expression between HMPOS-2.5 and HMPOS-10 may be due to HMPOS-2.5 being in an earlier state of EMT or an EMT-like process than HMPOS-10. There are fewer pathways associated with the gene expression patterns in HMPOS-2.5 compared with HMPOS-10. Many of the upregulated genes in HMPOS-2.5 are associated with EMT, but only implicate only the role of TGF-β, Wnt/β-catenin, and EGFR/Akt pathways. The downregulation of *PIEZO2*, *IGFBP2*, and *ACVR2A* may be due to HMPOS-2.5 being at an early stage of an EMT-like process or may reflect the effectiveness of other mechanisms of chemotherapy resistance.

### 4.4. Cell Populations Variating from the Main Cell Lines

There were small populations of cells from each identity that clustered separately from the main cell lines HMPOS, HMPOS-2.5, and HMPOS-10. These clusters were 8, 10, 11, and 13, which were renamed HMPOS-Var-1, HMPOS-2.5-Var, HMPOS-10-Var, and HMPOS-Var-2, respectively, to distinguish the cell lines that comprised these clusters. These clusters were compared with the main cell lines that they were split from, though HMPOS-Var-2 was excluded from further analysis because the cell count of 249 was too low to obtain meaningful results. Clusters 1 and 5 were also compared because of the distinction of these clusters within the HMPOS-10 cell line in comparison with the homogeneity of the HMPOS and HMPOS-2.5 cell lines.

*LGALS3* and *THBS1* were downregulated in HMPOS-Var-1 when compared with HMPOS. The upregulation of these two genes was shown to be correlated with chemotherapy resistance when comparing the expression of HMPOS, HMPOS-2.5, and HMPOS-10. *LGALS3* and *THBS1* are both associated with the promotion of tumorigenesis, metastasis, and EMT [34,35,36,77,78]. *FSIP1* and *OMD* were upregulated in HMPOS-2.5-Var compared with HMPOS-2.5, which aligns with the pattern already seen for the upregulation of *FSIP1* and *OMD* in HMPOS-2.5 in comparison with HMPOS. These two genes are related to docetaxel resistance and the regulation of osteoblast differentiation, respectively [85,88]. In HMPOS-10-Var, *SFRP2* and *COL12A1* were downregulated, while *FSIP1* was upregulated in comparison with HMPOS-10. *SFRP2* and another collagen, *COL5A2*, were previously seen to be upregulated in HMPOS-2.5 in comparison with HMPOS. *SFRP2* controls the Wnt/β-catenin pathway and its upregulation promotes EMT, the upregulation of *COL12A1* and other collagen genes is associated with chemotherapy resistance, and the upregulation of *FSIP1* promotes chemotherapy resistance [79,80,81,85,102]. The existence of these small variant clusters from the large, homogenous main clusters for each cell line may have been the result of the divergence of each cell line as chemotherapy resistance evolved. 

Clusters 1 and 5 within the main HMPOS-10 cluster were also compared using IPA and revealed the inhibition of the EIF2 signaling pathway alongside the promotion of the mTOR signaling pathway and the sirtuin signaling pathway (Figure 5D). EIF2 signaling pathway upregulation promotes tumorigenesis, metastasis, and tumor hypoxia, which results in chemotherapy resistance [103]. Oxygen is necessary for DNA damage to occur with radiation and activate chemotherapeutic agents [103]. The mTOR signaling pathway regulates cell proliferation and apoptosis, and its upregulation is associated with tumorigenesis and chemotherapy resistance [104]. DNA repair, apoptosis, and drug metastasis are controlled by the sirtuin signaling pathway and its upregulation is associated with tumorigenesis, metastasis, and drug resistance [105]. The differences in the promotion and inhibition of these pathways between these two clusters suggest that there are competing strategies for chemotherapy resistance. This is particularly because the pattern observed in cluster 5 of the downregulation of the EIF2 signaling pathway, the upregulation of the mTOR signaling pathway, and the increased upregulation of the sirtuin signaling pathway in contrast to cluster 1 was distinctively different to the pattern seen in HMPOS-2.5 compared with HMPOS.

### 4.5. Overlap in Proteomics and Transcriptomics

In our previous paper investigating the proteomics of these HMPOS-derived cell lines and their exosomes, there were several genes that were expressed in similar patterns to the transcriptomic work performed [17]. *FSTL1* (follistatin-related protein 1) was upregulated in both HMPOS-2.5 and HMPOS-10 in the previous proteomic data and the scRNAseq results [17]. *GLUL* (glutamine synthetase) and *ENO2* (*γ*-enolase) were only upregulated in HMPOS-10 in parallel to the proteomics results [17]. *CDH2* (N-Cadherin) was upregulated in both HMPOS-2.5 and HMPOS-10 in the proteomics study, but only in the HMPOS-10 transcriptomics [17]. *FSTL1* is a marker of EMT and invasion, *GLUL* plays a role in glutamine metabolism, *ENO2* increases glycolysis, and *CDH2* is a marker of EMT [17,106,107,108,109].

CTNNB1 (β-catenin) was seen to be upregulated in the previous proteomic analysis in HMPOS-10 and in the naïve HMPOS cell line when treated with exosomes derived from the chemoresistant cell lines [17]. These results and the difference in the expression of dephosphorylated and phosphorylated β-catenin between cell lines indicated the importance of CTNNB1 in chemotherapy resistance [17]. The scRNAseq results showed a downregulation of *CTNNB1* in HMPOS-10 in comparison with HMPOS and HMPOS-2.5, which is not in line with the proteomic results [17]. As has been recurrently documented in proteome–transcriptome integration approaches, these data do not always align. The transient nature of the expressed genes tends to be more dynamic, and the proteome more static [110,111]. Future multiomic approaches provide promise for better understanding the covariation in these mechanisms [112]. However, the observed upregulation of *SFRP2* and *COL5A2* in HMPOS-2.5 and *S100A6* and *TPM2* in HMPOS-10 indicates the importance of the Wnt/β-catenin signaling pathway in chemoresistance [17,56,58,79,91].

## 5. Conclusions

There are clear distinctions in the stage of the EMT-like phenotype and the mechanisms of tumorigenesis and chemotherapy resistance between HMPOS, HMPOS-2.5, and HMPOS-10. The differentiation of HMPOS-2.5 and HMPOS-10 from HMPOS demonstrates the ability of cancer cells to acquire resistance when under selection pressure from exposure to chemotherapy drugs. The contrast between the mechanisms of the HMPOS-2.5 and HMPOS-10 cell lines shows the complexity of chemotherapy resistance and the evolution of the adaptative mechanisms of cancer cells. The investigation of subclonal populations of each cell line helps to explore to evolution and acquisition of chemotherapy resistance. The EIF2 signaling pathway, the mTOR pathway, and the sirtuin pathway particularly seem to play important roles in chemotherapy resistance and the differentiation of HMPOS-10. The sirtuin pathway is known to play an essential role in maintaining malignancy, affecting cell longevity [113]. The role of sirtuins is complex and varies between cancer types, though evidence suggests that sirtuins have an inhibitory effect on cell viability. Interestingly, the member *SIRT1* induces EMT and enhances prostate cancer cell migration and metastasis [114]. The gene expression patterns of each cell line indicate the correlation of the EMT-like phenotype with chemotherapy resistance. Fifteen genes that were upregulated and five genes that were downregulated were strongly associated with EMT in the previous literature. Nine of these genes, *TMSB4X*, *LGALS3, COL3A1*, *HMGA1, S100A6*, *TPM2*, *CCND2*, *TFPI2*, and *HMGA2*, were either upregulated or only expressed in HMPOS-10. The previous proteomic analysis performed using these HMPOS-derived cell lines showed that the EMT marker *FSTL1* was upregulated in both HMPOS-2.5 and HMPOS-10, as previously observed, but *CDH2* was only upregulated in HMPOS-10 in this experiment. This indicates that HMPOS-10 is further involved in the EMT-like process in comparison with HMPOS-2.5.

This study was only able to evaluate the transcriptome of each cell and does not provide a full picture of the genetic landscape between these cell lines, nor the cell surface protein complement. This experiment also does not encapsulate the effects of the tumor microenvironment or transmissible chemotherapy resistance. A lack of correlation in expression between the previous proteomics and the present transcriptomics data may be due to annotation differences. The genetic divergence of these three cell lines will be explored in the future, along with the potential causative mutations that are affecting the regulation of genes associated with these varying levels of chemotherapy resistance. Another limitation of this study is the use of a long-established cell line, which may mean that the results do not fully reflect the mechanisms involved in vivo, though it allowed us to clearly see the genetic divergence from a very homogenous cell line. This can be rectified in future investigations by developing primary cell lines from tumor samples that have and have not been treated with chemotherapy to compare the expression of the tumor pre- and post-treatment. Additionally, comparisons of the tumor microenvironment, bone marrow distal to the tumor, and healthy bone marrow could be utilized for expression analysis using the methods in this work. Examining transcriptomics at the time of excision between cell populations of the heterogeneous tumor tissue and homogeneous cell lines would add to future experimental designs in the context of conveying chemotherapy drug resistance.

These results indicate the importance of the EMT-like process in the evolution of chemotherapy resistance. This analysis provides insight into potential treatment targets and illustrates the importance of accounting for the tumor evolution trajectory at the start of and over the progression of treatment to combat chemotherapy resistance in OSA.

## Figures and Tables

**Figure 2 genes-14-00558-f002:**
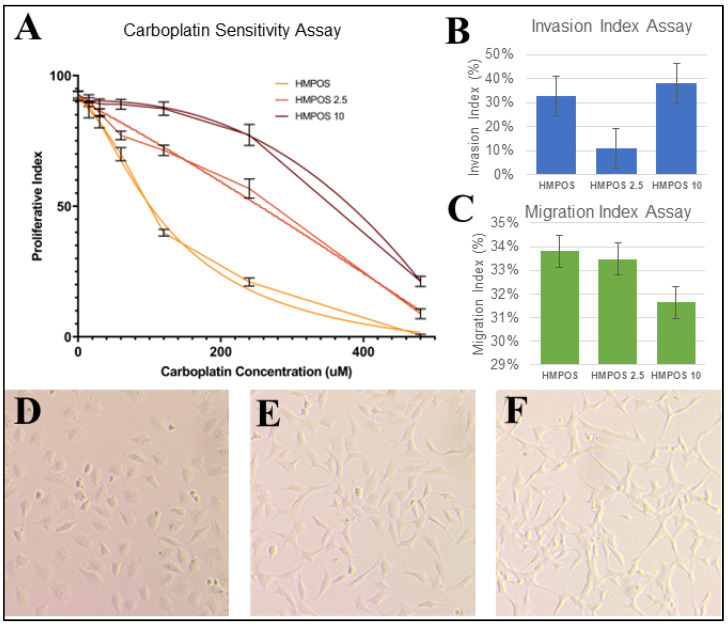
(**A**) Proliferative index for each HMPOS cell line at carboplatin dosages of 0 μM, 15 μM, 30 μM, 60 μM, 120 μM, 240 μM, and 480 μM. Three technical replicates of three biological replicates are shown. Nonlinear regression curve for resistance to carboplatin is significant according to the extra sum-of-squares F (*p* < 0.001). (**B**) Average % of Invasion Index Assay; the *p*-value was significant and found to be 0.00013753 using an ANOVA single-factor test. However, the Tukey’s HSD *p*-values between HMPOS and HMPOS-2.5 and between HMPOS-2.5 and HMPOS-10 were 0.001 and 0.001 respectively, while it was 0.379 between HMPOS and HMPOS-10. (**C**) Average % of Migration Index Assay; the *p*-value was not significant and found to be 0.06012698 using an ANOVA single-factor test. Light microscopy photos of (**D**) HMPOS, (**E**) HMPOS-2.5, and (**F**) HMPOS-10 showing distinct phenotypic differences, from cuboidal to spindle morphology.

**Figure 3 genes-14-00558-f003:**
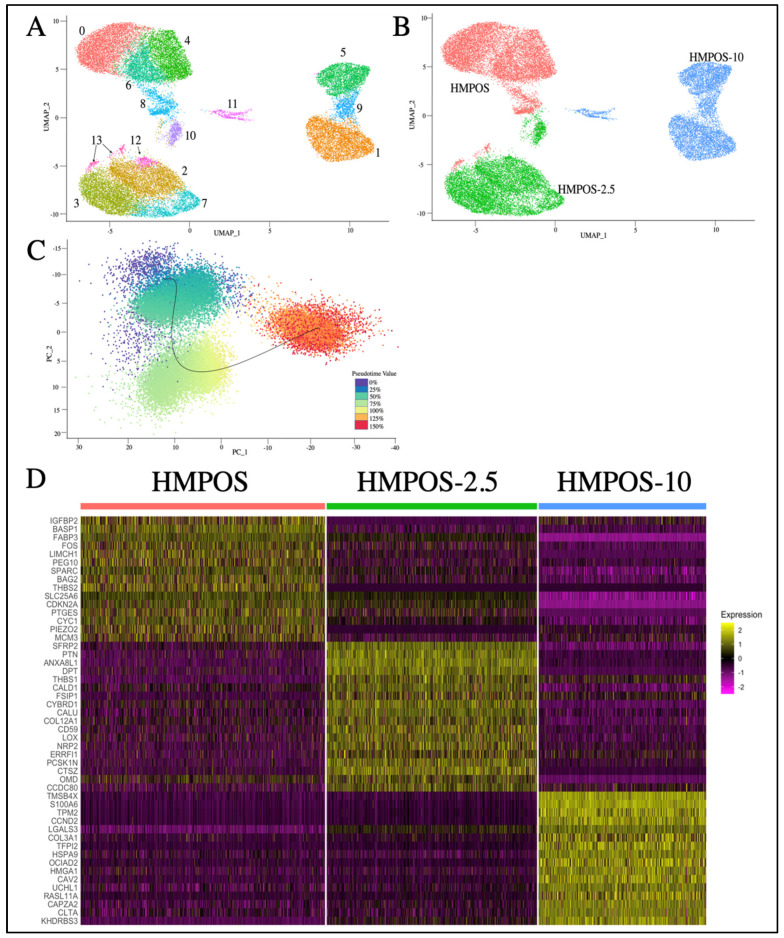
(**A**) UMAP plot of HMPOS, HMPOS 2.5, and HMPOS 10 showing the cell lines. (**B**) UMAP of the Seurat clusters in which HMPOS makes up clusters 0, 4, 6, 8, and 13, while HMPOS-2.5 forms clusters 2, 3, 7, 10, and 12, and clusters 1, 5, 9, and 11 are made up of HMPOS-10 cells. (**C**) PCA jitter plot highlighting the different cell stages and inferred trajectory. (**D**) Heatmap showing the highest differentially expressed genes for each cell line. This list was put together by selecting the top 20 genes for each cell line based on avg_log2FC, which is the log fold-change in the difference in expression between groups where positive values indicate that the expression of a gene is greater in the first group. Eight genes with no gene symbol listed were dropped.

**Figure 4 genes-14-00558-f004:**
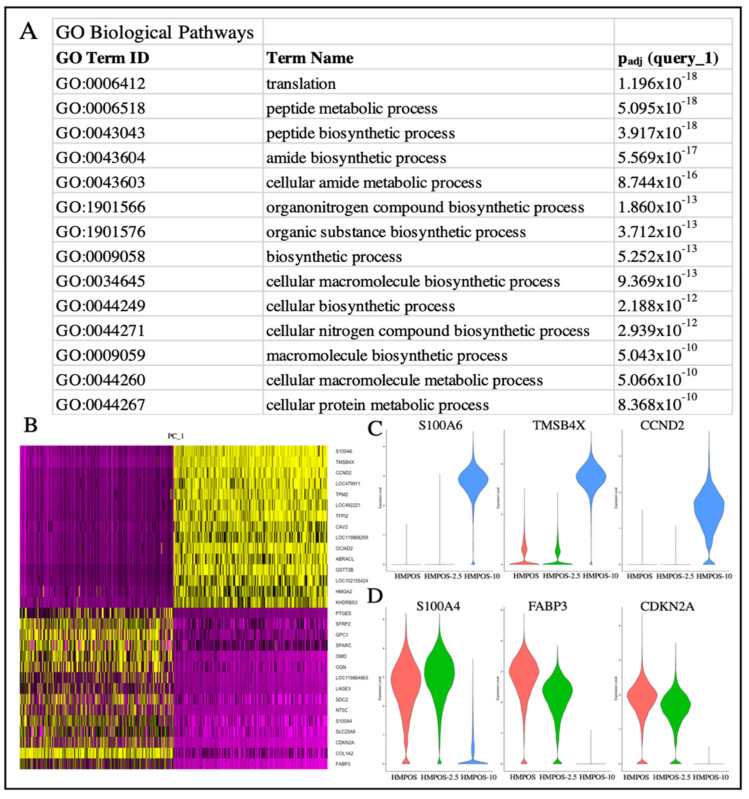
(**A**) gProfiler analysis of the biological pathways related to the upregulated genes of the HMPOS cell line in comparison with the 2.5 and HMPOS-10 cell lines. (**B**) DimHeatmap showing the primary sources of heterogeneity. (**C**) Violin plot of genes from PC_1, i.e., the most highly differentially expressed genes that coordinate with the HMPOS-10 cell line. (**D**) Violin plot of genes from PC_1, i.e., the most highly differentially expressed genes that coordinate with the HMPOS-2.5 and HMPOS cell lines.

**Figure 5 genes-14-00558-f005:**
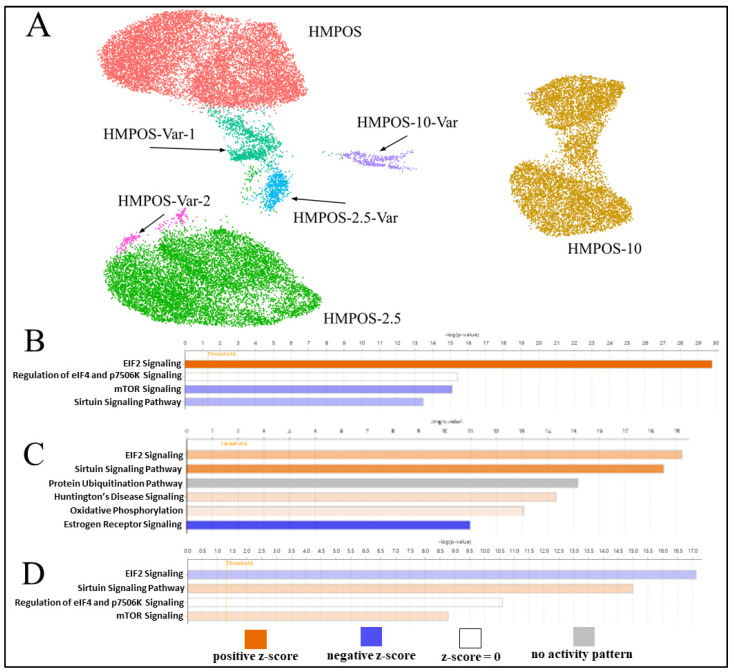
(**A**) UMAP plot of HMPOS, HMPOS-2.5, and HMPOS-10 and the subclonal populations that were distinct from the main cell lines. HMPOS-Var-1 and HMPOS-Var-2 are variants of HMPOS, though the low cell count of 249 cells of HMPOS-Var-2 led to its exclusion from further analysis. HMPOS-2.5-Var is a variant of HMPOS-2.5 and HMPOS-10-Var is a variant of HMPOS-10. (**B**) Pathway analysis comparing HMPOS and HMPOS-2.5, with positive z-scores in orange indicating pathway activation and blue indicating pathway inhibition in HMPOS-2.5 compared with HMPOS. The color’s intensity represents the z score’s distance from the mean, with darker color indicating greater distance. (**C**) IPA comparing HMPOS-10 with HMPOS and HMPOS-2.5. (**D**) IPA comparing HMPOS-10 cells in clusters 1 and 5, which can be seen in Figure 3B.

## Data Availability

All raw sequencing data have been submitted to the NCBI Sequence Read Archive (SRA) under BioProject PRJNA934861. The codes used in all analyses are available at www.github.com/EvoMedLab/HMPOS/.

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
