# Peer review of "Cellular Transcriptomics of Carboplatin Resistance in a Metastatic Canine Osteosarcoma Cell Line"

_genes, 2023, doi:10.3390/genes14030558_

Round 1
Reviewer 1 Report
Overall, an interesting study that presents data important for future work in OSA. Manuscript can use some editing work, especially in the introduction, with references, and the supplemental figures need a complete redo in both numbering and spell check, editing, etc. The conclusion overreaches and focuses mainly on EMT, but it would be really nice to instead focus on the genetic diversity seen and how this contributes to chemotherapy resistance - step away from EMT terminology (morphology change is interesting and important - a tumor that starts as mesenchymal technically doesn't go through EMT) and focus on the individual genetic changes. If a change is not significant, do not try to make it sound important with "trending to significance" or stating, "but the error bars don't overlap" and instead state WHY they might not be significant. And HOW can you use this data in future studies? Why is this important (and I think it is, I just want you to tell me why)? Lots of potential here!
Abstract/discussion: EMT versus "EMT-like" terminology is debatable; OSA is a mesenchymal tumor to start so "epithelial-to-mesenchymal" transition doesn't actually make sense, although it is often used in the literature. This is a clarification that is editor/journal dependent.
Introduction 2nd paragraph discusses human OSA and references 6-8 are canine references - please pull and references for human disease or clarify that implant/injury refers to canine OSA.
Introduction is quite lengthy and has many references - consider condensing the introduction to make it more succinct and a relevant introduction to the work you present - this felt like a very brief review of all of OSA. Focus instead on the points that relate to your research. For example, how relevant is the age of diagnosis, impact of injury, breed, implants, etc. on YOUR study?
3.1 results: "This trend was found to not be statistically significant with a p-value of 0.267, but the standard error bars of HMPOS do not overlap with HMPOS-2.5 and HMPOS-10 at 120 ?? and 240 ??." No standard error bars present on graph, and it is unclear what this statement means. Does the lack of overlap of standard error bars that we can't see indicate significance when the statistics don't? Please clarify.
Supplementary figures are incorrect. Example S1 is a carboplatin sensitivity figure, in your paper S1 refers to distribution of cell phases.
Conclusion: Only HMPOS cell lines used - the use of other cell lines and how this might benefit future study not discussed. Completion of EMT by HMPOS-10 is a bit of a reach for the conclusion - can't quite say that form the data presented.
Reviewer 2 Report
General Comments
The authors present data as a follow up to a previous publication in which the HMPOS cell line was exposed to carboplatin in culture to establish carboplatin resistant cell lines, HMPOS-2.5 and HMPOS-10. While the previous publication characterized these cell lines for the sensitivity to carboplatin and proteomic profile, the current manuscript presents additional data regarding carboplatin sensitivity and characterizes their transcriptome using single cell RNAseq.
Despite the proteomic data from the author’s previous publication showing increased expression of several EMT markers in the carbo resistant cell lines, that was largely ignored in the previous manuscript’s discussion. The authors now focus on the observed EMT in the carbo resistant HMPOS cell lines which, similar to the proteomic data, is supported by the new RNAseq data. However, the authors make no comparisons between the previous proteomic data and the current RNAseq data, and instead introduce their findings regarding EMT as novel. The observed EMT induction, however, was already evident from the previous manuscript’s data.
While possibly not the author’s intention, the current manuscript appears to ignore all the characterization data presented in their previous manuscript for these same HMPOS cell lines, and consequently presents the current data as more novel than it is. This is evident in the text and language used in the current manuscript which implies (unintentionally) that some data is novel, despite having been previously reported. The text and wording throughout the manuscript should be carefully reviewed to avoid this. Furthermore, the manuscript would benefit greatly from a more comparative analysis of the previously published proteomic data and the new RNAseq data.
In addition, several citations appear to be either misnumbered or not appropriate for supporting the text. Statistical analysis for several experiments is either not explained in the methods (such as unknown controls used), inappropriate for the data, or missing from the results.
Specific Comments:
Abstract
The language the authors use to introduce the HMPOS, HMPOS-2.5, and HMPOS-10 cell lines in the abstract is misleading. The authors mention “Two resistant cell lines and a naive cell line underwent single cell RNA sequencing…” The wording the authors use suggest 3 unique cell lines. I suggest the wording be altered to clarify that all of the “cell lines” used here were derived from the HMPOS cell line.
Methods
2.1 Generation of induced carboplatin chemoresistant cell lines
The subtitle here implies that the “generation” of the chemoresistant HMPOS cell lines is novel here, but they were already generated and previous characterized (ref #10). The authors should change the wording to reflect this. Perhaps remove “Generation of induced”.
When describing the HMPOS cell line, the authors write “This cell line was derived from the canine OSA cell line POS, which was generated by harvesting metastases cells from mice injected with canine OSA cells.” The wording suggests that the parent POS cell line was derived from metastatic lesions passaged in mice, rather than the HMPOS cell line being derived from metastatic lesions in POS injected mice.
It’s not clear how many passages the different HMPOS cell lines have undergone since there generation following carboplatin treatment. The author’s previous publication mentions “10 generations”- is that similar to passages? Regardless, how many passages have each cell line undergone since the initiation of carboplatin sensitivity for the current study? With each subsequent passage, the cell lines become more susceptible to genetic drift, which can cause gene expression changes not specific to the initial carboplatin sensitivity.
2.2 Carboplatin sensitivity assay.
The authors use non parametric one-way ANOVA (Kruskal Wallis) for statistical analysis of dose response curves, which is not recommended and not meaningful. They essentially compare responses at each carboplatin concentration, rather than the overall response to treatment. Non-linear regression should be used to determine IC50 values and curve fit analysis used to compare responses between cell lines.
2.3 Cell invasion Assay.
“Thincerts were placed in a 24well plate and coated with coating buffer. “ – Do the Authors mean “coated with Matrigel Matrix”?
There are a lot of details here that could be consolidated. While I agree that more details are generally helpful, this type of Invasion assay is standard in practice and therefore all the details are not necessary, unless a significant deviation to the traditional assay was used.
The authors should more clearly describe how they distinguish between “migration” and “invasion”. Although this could be implied from the different index formulas, it would be helpful to point out that it is the presence of the Matrigel Matrix that is important. Essentially the positive control wells without the Matrigel are measuring migration, while the experimental wells with the Matrigel are measuring invasion. In addition to being the positive control for the invasion group, the matrigel-free + serum wells are also a second experimental group measuring migration. The authors should clarify this in the methods.
What statistical analysis was used to compare invasion and migration ability between the cell lines?
2.4 Single cell RNA sequencing
The authors write “Reverse transcription was then performed on the RNA barcodes of each GEM.” What do the authors mean by this? The reverse transcription is performed on the RNA and the barcodes are added during the reverse transcription reaction for this 10X genomics Chromium Single Cell kit. The barcodes are part of the primers used. The authors already state that the Chromium Single Cell 3’ reagent kits were used according to the manufacturer’s instructions, and therefore these details regarding the protocol are not necessary.
Where all the single cell RNAseq protocols performed by the authors themselves, or was a core facility or commercial vendor used?
2.5 Analysis and visualization
The authors use the canine build “UU_CanFam_GSD_1.0” reference for alignment of the RNAseq data. The more common and recognized name for this build is “camFam4”. I suggest changing the text to read “UU_CanFam_GSD_1.0/canFam4” or just “canFam4”.
The methods here include some very specific details while omitting more meaningful information. For example, the authors write: “Raw base call files (BCL) were demultiplexed using the “mkfastq” command to generate FASTQ files”. Where did the authors use this command? Did the authors use a software such as bcl2fastq conversion software? If so, they should simply state that “bcl2fastq conversion software was used to generate FASTQ files.” A bioinformatics expert should review the text of the methods.
The references appear to be off here. For example, “Gene set enrichment analysis was performed on the upregulated and downregulated gene lists comparing HMPOS to HMPOS-2.5, HMPOS to HMPOS-10, and HMPOS-2.5 to HMPOS-10 in gProfiler [33].” Ref 33 is a review paper about EMT and nothing to do with gene set enrichment using gProfiler. The authors should perform a thorough review of their references.
What significance threshold (False Discovery Rate) was used for the various analysis? For enrichment analysis, were the included DEGs based on p-values, or Padj.?
Single Cell RNAseq would typically identify individual cell populations within a sample, but given the samples used here are essentially clonal cell populations, it’s not clear how the samples were grouped for statistical analysis. The authors present data showing the different cluster groups within of the HMPOS cell lines (#1-13), but there does not appear to be any analysis comparing the clusters within each cell line. Was each cluster within a cell line considered a technical or biological replicate for that cell line when performing DE analysis? Or were the clusters for each cell line pooled together as an individual sample for analysis? If so, what was the purpose of doing the single cell RNAseq? It appears the methods used here resulted in extremely exaggerated significance values, as evident in the Padj. values in Figure 4A for gene ontology. It seems standard bulk RNAseq could have been performed with similar results, but with higher p-values, depending on the number of technical replicates used.
Why was single cell RNAseq used instead of bulk RNAseq?
Results
3.1 Characterization of the HMPOS cell lines.
These HMPOS cell lines were already characterized in the authors previous publication (ref #10).
“The correlated resistance of HMPOS-2.5 and HMPOS-10 in comparison to HMPOS was validated with cell viability assay (Figure 2).” – The authors already validated this in their previous manuscript (Ref #10) using an MTS proliferation assay. Although different results were obtained, the trend regarding carboplatin sensitivity was similar between studies and therefore the current data provides confirmation of previous data, not validation of each cell line’s sensitivity to carboplatin.
How do the results in the carboplatin sensitivity assay presented here compare to the carboplatin sensitivity assay previously described by the authors in reference #10? It appears the HMPOS and HMPOS-2.5 cell lines have developed tolerance to carboplatin since the previous publication, while the HMPOS-10 cell line has become much more sensitive. In the author’s previous manuscript, both the HMPOS (niave, S) and HMPOS-2.5 cell lines had less than 40% “proliferative index” at the 50uM dose after 72hrs of treatment. I assume “proliferative index” is similar to % viability, or % relative to control, but the methods for calculating “proliferative index” are missing from the previous publication. In the current publication, the 50uM concentration of carboplatin with 72hr exposure seems to have very little effect on these cell lines. However, it is also not clear what is meant by “Condition Average (%)”, there is nothing in the current methods that describes how “Condition Average (%)” was calculated or how it relates to the “cell viability” measured by the Cellometer Auto 2000. Similarly, the HMPOS-10 cell line was completely resistant to carboplatin up to about 1mM concentration in the previous publication (which is quite remarkable with a 72hr drug exposure), but in the current manuscript, the same HMPOS-10 cell line has about 20 “Condition Average (%)”. Assuming the “Condition Average %” is the similar to “proliferative index” and both indicate % viability as compared to an unspecified control, can the authors explain the remarkable difference in response to carboplatin between the two studies?
Figure 2A – what is the X axis? What is “condition average (%)”? As mentioned above, this appears to be percent survival/viability, relative to a control, but it is not clear what that control is. This is not explained in the methods and the authors need to elaborate here.
Figure 2A – The methods describe statistical comparison, but there is no indication of significance presented in this figure.
Invasion assay results – why is this data in the supplementary files? A small figure with a bar graph comparing the invasion and migration index scores of the cell lines would be appropriate for the main text.
3.2 Distinction in transcriptomes…
The authors write “HMPOS makes up clusters 0, 4, 6, 8, and 13 while HMPOS-2.5 clusters out to 2, 3, 7, 10, and 12, and clusters 1, 5, 9, and 11 are made up of HMPOS-10 cells (Figure 3B).” This identifying information should be in the figure legend, not the text of the manuscript. It has no meaning outside the figure.
Figure 3 - What is meant by “original identities” here? For 3D, the figure legend implies that the “original identities” are the 3 HMPOS cell lines (control, 0.25, and 1.0). But in 3B, the legend implies the “original identities” are represented by the unique clusters (#1-13). Which is it? This is very confusing.
Figure 3D - The methods for generating this heat map are unclear. It appears the authors generated a heat map using only the top 60-80 genes that were differentially expressed between groups (all groups?), is that accurate? If so, were the genes selected based only on fold change in expression, or based on the most significantly differentiated genes? The authors need to elaborate on the methods used here. What significance threshold was used?
Also Figure 3D – the “Identity” information next to the heat map does not match the names above the heat map. I suggest removing that side legend as it is redundant. Referring to the cell lines as “cell lines” rather than “identities” would also avoid confusion.
Discussion
The authors write: “Cell viability assay confirmed the induction of carboplatin drug resistance with HMPOS being the most sensitive, HMPOS-2.5 having improved survival, and HMPOS-10 exhibiting the most resistance to carboplatin exposure.” – The authors should cite their previous publication here (#10) as they indicate these results are “confirming” those previous results (although some significant discrepancies between the results of both studies).
The authors write: “The observed morphology changed from cuboidal to spindled correlates with the morphology changes seen in the epithelial-to-mesenchymal transition (EMT) where spindled tumor cells are more aggressive and chemoresistant [34].” Reference 34 describes TMSB4X as a prognostic marker in Head and Neck Squamous Cell Carcinoma, with only a brief mention of potential relationship between TMSB4X and EMT in the discussion. This is not an appropriate reference for the author’s statement regarding their current data and EMT.
The author’s previous publication with these HMPOS cell lines (Ref#10) included a full proteomic analysis of each cell line. However, the authors do not make any reference to that data in the current manuscript, nor do they provide any correlative analysis between the proteomic data and the RNAseq data. This comparison is logical and clearly relevant. Why is this not included in the discussion?
On that note, when comparing the data between the 2 studies, there does not appear to be much overlap in expression with regard to specific genes and proteins, although clear overlap in pathways is present (EMT regulation, for example). One reason for this is likely annotation differences between the genome build used to align the RNAseq (CamFam4.0) and the Protein database used in the previous publication. For example, S100A7 was identified in the protein analysis, but not the RNAseq analysis. However, S100A7 is not annotated in CanFam4.0 and therefore any reads aligning to S100A7 would have been ignored, which is likely why it was not identified as DE in the RNAseq data. On the other hand, S100A6, which was identified as DE the RNASeq data, is annotated in CamFam4.0. Interestingly, the GSTT2B pseudogene is the only GSTT related gene annotated in CamFam4.0, and it is not annotated in any other canine genome build (Canfam3, 5, or 6). In fact, none of the “real” canine GSTT genes are annotated in any of canine genome builds. As such, had the authors aligned to anything other than CanFam4, DE of GSTT2B would not have been identified.
I don't mean to imply that all of that should be in your discussion, but at the very least, a basic discussion of the differences and similarities between the Protein and RNA data sets of the two studies is warranted and would add value to your discussion.
Finally, the authors ignore a major limitation of this study, which is that all the presented data comes from analysis of a single cell line, HMPOS. This cell line has been around for a long time, passaged many, many times, and established from another cell line after several passages in mice. As such, the observed results may be limited to HMPOS and not reflect the mechanisms involved in chemoresistance in osteosarcomas within the larger canine patient population. The authors should address this in their discussion.
Round 2
Reviewer 2 Report
The authors have made several improvements to this manuscript and have appropriately addressed the majority of my comments. However, I do have a few additional, minor comments that should be addressed prior to publication.
Author associations – “3” and “4” associations are not assigned to any of the authors.
Introduction -
“Though similar genes are involved, the specific mutations are infrequently shared between species with the exception of TP53 and SETD2 and DMD are unique to canine OSA [6,8–10].” - There appears to be an editing error with this sentence. Do the authors mean “Though similar genes are involved, the specific mutations are infrequently shared between species with the exception of TP53 [insert refs]. Mutations present in SETD2 and DMD are unique to canine OSA [6,8–10].” – or something similar?
Results -
Proliferation assay. IC50 values were calculated, but the statistical analysis used to compare response curves between the cell lines is unclear. Although there appear to be differences in the IC50 values, the authors need to show those differences are statistically significant. This can be accomplished by comparing the best fit values between two curves using extra sum of squares F Test.
Figure 2B – Given that the ANOVA shows significance (p=0.00013753), multiple comparisons should be performed here – Tukeys, or appropriate. ANOVA is showing the means are different, but it looks like the HMPOS 2.5 is the only one different from the HMPOS. This seems to suggest at least some gene expression changes observed in the HMPOS-2.5 influence invasion, but they may be exclusive of those influencing chemoresistance. Regardless, additional studies (outside the scope of the current study) would be required to evaluate that interaction.
Figure 5 B, C, and D – The text on the Y axis is very difficult to read. Larger font or a higher quality image is needed. In addition, what do the different shades of orange and blue indicate? Is that simply to distinguish the different pathways in each graph? If so, I don't think that is necessary – each bar is labeled along the Y axis, and therefore the different shades are redundant and only adds confusion, in my opinion.
Discussion –
“The invasion and migration assay results did not reach significance, but are consistent with the HMPOS cell line being a highly metastatic line of POS cells that are homogenous prior to carboplatin resistance.” The ANOVA does show a significant difference between the 3 groups. This should be addressed in the discussion.
Page 10 – 3rd paragraph – “downregulates miR-381 expression to regulate EMT [38,39]].” – remove extra “]”.
Conclusion -
Last paragraph - “This can be rectified in future investigation by developing primary cell lines from tumor samples that have and have not been subjected to chemotherapy. to compare expression of the tumor pre- and post-treatment.” – is that first period a typo?
“CTNNB1 (β-catenin) was seen to be upregulated in the previous proteomic analysis in HMPOS-10 and in the naïve HMPOS cell line when treated with exosomes derived from the chemoresistant cell lines [17]. These results and the difference in expression of dephosphorylated and phosphorylated β-catenin between cell lines indicated the importance of CTNNB1 in chemotherapy resistance [17]. The scRNAseq results showed a downregulation of CTNNB1 in HMPOS-10 in comparison with HMPOS and HMPOS-2.5, not in line with the proteomic results [17].”
There are many potential explanations for the differences in B-catenin protein and RNA expression in the HMPOS-10 cell line, compared to HMPOS and HMPOS-2.5. It is possible that the B-catenin protein has become more stable or resistant to proteolysis in the HMPOS-10 cell line, potentially over activating a negative feedback loop, resulting in downregulation of the RNA, while maintaining high protein levels. Assuming the protein is still functional, the overall effect would be increased B-catenin activity. Discrepancies between RNA and Protein expression are common in cancers and are not necessarily contradictory.
Aside from the minor revisions mentioned above, the authors have adequately addressed all of my other concerns and comments. I commend the authors on this well written and informative manuscript and thank them for this valuable contribution to veterinary oncology research.
